# Impact of Nutritional Status on Total Brain Tissue Volumes in Preterm Infants

**DOI:** 10.3390/children11010121

**Published:** 2024-01-18

**Authors:** Cyndi Valdes, Parvathi Nataraj, Katherine Kisilewicz, Ashley Simenson, Gabriela Leon, Dahyun Kang, Dai Nguyen, Livia Sura, Nikolay Bliznyuk, Michael Weiss

**Affiliations:** 1Division of Neonatology, Department of Pediatrics, University of Florida, Gainesville, FL 32608, USA; cndvaldes@gmail.com (C.V.); parvathinataraj@yahoo.com (P.N.); kisilk@ufl.edu (K.K.); livia.sura@peds.ufl.edu (L.S.); 2College of Medicine, Gainesville Campus, University of Florida, Gainesville, FL 32608, USA; ashleysimenson@ufl.edu (A.S.); g.leon@ufl.edu (G.L.); dahyun.kang@ufl.edu (D.K.); 3Department of Pediatrics, University of Florida, Gainesville, FL 32608, USA; dainguyen@ufl.edu; 4Department of Agricultural & Biological Engineering, University of Florida, Gainesville, FL 32608, USA; nbliznyuk@ufl.edu

**Keywords:** neonatal brain, brain tissue volume, neonatal nutrition, preterm, malnutrition, weight z-score, length z-score

## Abstract

Preterm infants bypass the crucial in utero period of brain development and are at increased risk of malnutrition. We aimed to determine if their nutritional status is associated with brain tissue volumes at term equivalent age (TEA), applying recently published malnutrition guidelines for preterm infants. We performed a single center retrospective chart review of 198 infants < 30 weeks’ gestation between 2018 and 2021. We primarily analyzed the relationship between the manually obtained neonatal MR-based brain tissue volumes with the maximum weight and length z-score. Significant positive linear associations between brain tissue volumes at TEA and weight and length z-scores were found (*p* < 0.05). Recommended nutrient intake for preterm infants is not routinely achieved despite efforts to optimize nutrition. Neonatal MR-based brain tissue volumes of preterm infants could serve as objective, quantitative and reproducible surrogate parameters of early brain development. Nutrition is a modifiable factor affecting neurodevelopment and these results could perhaps be used as reference data for future timely nutritional interventions to promote optimal brain volume.

## 1. Introduction

Nutritional status plays a substantial role in the development of premature infants. The fetal brain is a highly metabolic organ, and it consumes the greatest number of nutrient resources for its function and growth during the second and third trimester of gestation [1]. For the very premature infant, this crucial period of brain development and maturation will take place in the ex-utero environment (i.e., 24–40 weeks’ postmenstrual age), where there is increased risk of poor growth and malnutrition [2]. Undernutrition during this period negatively impacts rapidly growing brain regions [1,3]. The risk of malnutrition is due to several causes such as reduced nutrient stores at birth, immature nutrient absorption, organ immaturity, high levels of energy expenditure secondary to neonatal illness and delayed advancement of enteral feeds due to cautious feeding management protocols [4]. The dependence on health care providers to accurately identify and effectively provide needed nutrients during a period of rapid growth and development also plays an important role [5]. Poor brain growth in this critical period of development may be the result of inadequate early nutrition with consequential poor neurodevelopmental outcomes [2,6,7].

There is an increasing body of literature over the last 30 years investigating the correlation of brain volumes at term equivalent age with prematurity [1,2,3,8,9,10,11,12,13,14,15,16,17]. Fetal brain MRIs have shown that the greatest rate of growth occurs between 25 weeks and 40 weeks’ gestation [8,9]. There are several postulated non-nutritional factors that may alter how nutrients are accreted and distributed, contributing to brain volume structural changes in the premature population including intraventricular hemorrhage (IVH), white matter injury, infection, use of corticosteroids, presence of a hemodynamically significant PDA (*hs*PDA) and inflammation [1,10,11,18]. Evidence demonstrates up to 89% to 97% of infants have weight and length measurements below the 10th percentile at 36 weeks’ postmenstrual age, and most remain as such by 5 years of age [7,19]. Growth parameters such as change in weight gain and linear growth independent of weight gain are the recommended indicators of nutritional status [20]. While many factors influence neurodevelopmental outcomes in preterm infants, nutrition is a modifiable element that can be optimized by healthcare providers.

Preterm birth is associated with altered brain development, with younger gestational age (GA) at birth often associated with greater brain volume reduction [17,21]. To provide a representation of the brain structure, neonatal brain MRI atlases have been developed and are widely used to determine tissue contrast allowing for classification. Hüppi and colleagues [12] were among the first to document marked maturational changes in brain tissue volumes from 29 to 41 weeks post conceptional age, using a volumetric brain tissue segmentation technique. *T*_2_-weighted sequences provide optimal tissue contrast due to higher cell density in grey matter. For this reason, many neonatal imaging tools and studies employ *T*_2_-weighted images alone or in combination with other sequences [13].

In this study, we aimed to determine the associations between nutritional status of extremely premature infants and total brain tissue volume at term equivalent age (TEA), utilizing the maximum change in weight and length z-scores as nutritional indicators. We collected and analyzed data for variables pertaining to patient demographics, type, and percentage of enteral feed and other potential confounding variables such as maternal history, infant mechanical ventilation use, steroid use, comorbidities to include BPD, gastrointestinal injury, hemodynamically significant PDA, culture proven sepsis and continuous use of sedation. Utilizing recently published malnutrition guidelines for preterm infants, we hypothesized that brain volume reduction at TEA for preterm infants born at <30 weeks would correlate with the degree of malnutrition during their NICU treatment course.

## 2. Materials and Methods

### 2.1. Participants

The study was reviewed and approved by the Institutional Review Board at the University of Florida. The study design involved a retrospective chart review which included all infants born <30 weeks who were admitted to the neonatal intensive care unit of the UF Health Shands Children’s Hospital between the period of January 2018 and July 2021. Infants were excluded if they had chromosomal abnormalities, major gastrointestinal malformations, intracranial abnormalities such as colpocephaly, absent septum pellucidum, porencephalic cyst, encephalomalacia, etc., and those who did not survive to term equivalent age were excluded as well. Using the UF Health Vermont Oxford Network data, infants ranging in gestational ages from 22 to 29 weeks in the database were identified for analysis. In our unit, we routinely perform brain MRIs on infants born prior to 30 weeks, when they reach term equivalent age at 36 weeks postmenstrual age.

### 2.2. Data Collection

Demographic data including gestational age at birth, gender, size, length of hospitalization and birth weight were recorded. Infants had nutritional data documented weekly in the medical chart as part of their routine care during their admission. The data was originally calculated by an experienced neonatal registered dietitian using recommended neonatal malnutrition scores such as anthropometric measurements, maximum change in weight and length z-scores. Additionally, the percentage of enteral nutritional intake whether it was expressed breast milk, donor breast milk, and/or formula was recorded. Potential confounding variables including maternal factors such as preeclampsia, IUGR and antenatal use of steroids were collected. In addition, infant comorbidities such as bronchopulmonary dysplasia (BPD), days on mechanical invasive ventilation, presence of intraventricular hemorrhage (IVH), hemodynamically significant PDA, clinical sepsis, gastrointestinal injury, erythropoietin, sedation, and postnatal use were collected. BPD was defined as any respiratory support administered at 36 weeks’ postmenstrual age, regardless of supplemental oxygen use [22].

Criteria for a hemodynamically significant PDA was met when echo findings showed a PDA > 1.5 mm/kg and/or clinically the infant underwent surgical, medical treatment with acetaminophen and/or NSAIDS or conservative management with fluid restriction. Not all infants had echocardiography performed, however, if clinically they showed signs of ductal steal or pulmonary congestion, the infants received treatment and consequentially met the criteria to be included in this study. Clinical sepsis was defined as the presence of any positive blood culture during the infant’s hospitalization. Gastrointestinal injury was described as any history of necrotizing enterocolitis (NEC), spontaneous intestinal perforation (SIP) and/or short bowel syndrome. Erythropoietin used for anemia or as neuroprotector was included in the analysis. We also collected data regarding the number of days an infant received continuous sedation during their hospitalization and the number of sedatives received. The use of postnatal steroids used for more than one week, either hydrocortisone or dexamethasone, was also included in the analysis.

### 2.3. MRI Data Acquisition

Brain tissue volumes were measured by analyzing data from TEA brain MRIs obtained from a 1.5 Tesla MAGNETOM MRI (Avanto and Aera, Siemens Medical Solutions USA, Inc., Malvern, PA, USA) or a 3.0 Tesla MAGNETOM MRI (Vario, Prisma, and Skyra, Siemens, Munich, Germany). In our unit, brain MRIs without sedation or intravenous contrast are routinely obtained for all term equivalent age infants born <30 weeks’ gestation, thus no new images were specifically obtained for this study.

De-identified T2 weighted axial sequences were obtained and downloaded to ITK-SNAP software version 3.8.0, to perform manual segmentation and measure cerebral volumes in typical voxel size (mm^3^). Using the Mary Rutherford brain neonatal atlas to guide segmentations, different brain structures were delineated [23]. All supratentorial tissues were selected for analysis involving cerebral cortex, white and grey matter, basal ganglia, and thalami. These structures were added and computed as total brain tissue volumes. Ventricles, brainstem, cerebellum, and overt brain injuries were excluded. Each brain MRI was manually segmented twice by different investigators to ensure data quality and optimal 3D brain reconstruction as shown in Figure 1.

### 2.4. Nutritional Status

The indicators for preterm malnutrition used in our study were the maximum change in weight and length z-scores, based on the criteria published by Goldberg et al. [4]. In neonates with an uncomplicated postnatal adaptation, the change in weight z-score demonstrated a gain trajectory of 0.8 SD below birth at day of life 21 [24]. These indicators are collected weekly by the unit registered dietician, starting at week 2 of life and are documented in the infants’ electronic medical record. Anthropometrics such as weight are measured daily by bedside registered nurses and weekly for head circumference and length. Length is measured by placing the infant on their back in the center of a length board, measuring to the nearest 0.1 cm. To describe the different degrees of malnutrition, we utilized the criteria by Goldberg et al. [4], which defines both weight and length z-scores as mild malnutrition when there is a decline of 0.8–1.2 SD, moderate malnutrition as a decline > 1.2–2 SD, and severe malnutrition as a decline of >2 SD.

At the UF NICU, the nutrition guidelines are divided into parenteral and enteral managements. Parenteral nutrition (PN) starts on day 0 of life with total volumes (TV) depending on weight. If less than 1000 g TV = 80–100 mL/kg/day, if 1001–1750 g TV = 80–100 mL/kg/day and if greater than 1750 g, TV = 60–80 mL/kg/day. Lipids are started from day 0 of life at 3 g/kg/day and maxed protein at 3.5 g/kg/day. Initial GIR at 4–5 g/kg/min and titrated to a goal of 10–12 g/kg/min. Regarding electrolytes, sodium is generally not needed in the first 2–3 days of life, and potassium may or may not be included at 0–1 mEq/kg/day and withheld per renal function. Sodium or potassium acetate often is needed to correct metabolic acidosis in preterm infants. Calcium and phosphorus are started on day of life 0 as well, at 1–2 mmol/kg, with the amount needed based on serum values and renal function. Preterm infants are anticipated to be on PN for 1–2 weeks and started on small volume enteral/gavage feeds by day 0–1, if medically stable. Colostrum buccal priming is encouraged for infants < 1250 g, 0.2 mL every 6 h × 7 days. Feeding advancement is 10–30 mL/kg/day as tolerated. Maternal breastmilk is preferred, unless contraindicated. Donor breastmilk (DBM) and formula are also feeding alternatives. DBM is indicated for those infants < 30 weeks. Maternal breastmilk or DBM are fortified to 22 kcal/oz with a human milk fortifier at 60 mL/kg/day and to 24 kcal/oz at 100 mL/kg/day. If breastmilk is not available, special care formula 24 kcal/oz is given to the 30–34-week infants. If 34–36 weeks, Neosure 22 kcal/oz and for >37 weeks, Similac 19 kcal/oz.

### 2.5. Data Analysis

Classical normal theory linear regression models were used to explore the marginal associations between the response variables (e.g., brain volume) and the (quantitative and categorical) covariates of interest. In addition, we examined the conditional associations between the above covariates (X) while controlling for potential confounders (C). Specifically, since the effect of a confounder variable is not known to be additive, we evaluated both the additive models (with main effects X and C) and those with the main effects and the interaction of X and C. Model selection was performed by dropping the highest-order statistically nonsignificant terms sequentially [25].

## 3. Results

### 3.1. Subject Demographics and Characteristics

We identified 371 patients <30 weeks’ gestation, during the period of July 2018–July 2021 and 198 were included in the final analysis.



Baseline demographics and clinical characteristics are summarized as means and standard deviations in Table 1. The mean gestational age at which brain MRIs were done was 36 weeks + 5 days. We utilized the maximum change in weight and length z-scores as our primary malnutrition indicators. We found a mean decline in the group of −1.33 and −1.51, for weight and length z-score, respectively, both indicative of moderate malnutrition.

### 3.2. Relationship with Brain Tissue Volumes and Malnutrition

All infants in this cohort had some degree of malnutrition during their NICU stay. Maximum change in weight and length z-scores were found to have significant positive associations with brain tissue volumes at TEA (*p* < 0.05). In addition, length z-score is positively associated with brain volume in the various gestational ages (*p* < 0.05).

### 3.3. Association between Type of Enteral Feed and Brain Tissue Volumes

Many differences exist in nutrition practices, and they are constantly modified according to the medical necessities of each infant. Once infants transition from parenteral to enteral feeds, the amount and type of enteral feed (expressed breastmilk, donor breastmilk and/or formula*)* is measured routinely in percentages by the registered dietician in our unit. We analyzed the association between brain tissue volumes and type of enteral feed; however, there was no association found.

### 3.4. Intracranial Hemorrhage and Brain Tissue Volumes Association

In total, 109 (55%) of infants had no intraventricular hemorrhage (IVH) and 74 (37%) had low grade IVH, defined as the presence of either grade 1 or 2 IVH by Papile’s classification [26]. In total, 13 (6%) had high grade IVH, defined as the presence of either grade 3 or 4 [26]. And only 2 patients with periventricular leukomalacia (PVL) were identified. No significant associations were seen between intracranial hemorrhages or the presence of PVL and brain tissue volumes.

### 3.5. Demographics and Brain Tissue Volumes

Growth assessment begins at birth by identifying whether an infant is small for gestational age (SGA), appropriate for gestational age (AGA), or large for gestational age (LGA) or has experienced intrauterine growth restriction (IUGR). Linear associations with brain tissue volume at TEA and gestational age and size at birth were found (*p* < 0.05), but no differences were found between brain tissue volumes and gestational age alone. Smaller brain tissue volumes were significantly associated with the male gender. In addition, there was a significant positive association between birth weight and brain tissue volume (*p* < 0.05). (Figure 2 and Figure 3).

### 3.6. Associations between Multiple Clinical Variables and Brain Tissue Volumes

Neonatal comorbidities and maternal factors were analyzed to assess confounders. There was a significant negative association between brain tissue volumes and presence of BPD and clinical sepsis. No interactions were found between brain tissue volumes and maternal factors such as preeclampsia, antenatal steroids, or IUGR. Additionally, no associations were found between infant mechanical invasive ventilation, postnatal steroid and erythropoietin use, gastrointestinal injury, presence of a hemodynamically significant PDA or continuous sedation use.

## 4. Discussion

Throughout the last decade there has been an increasing survival of extremely premature infants. However, this survival is accompanied by an increasing awareness of subsequent neurodevelopmental deficits. To improve the outcome for these infants, it is crucial to understand both the nature of the cerebral structural abnormalities and any modifiable factor involved in its pathogenesis [21]. The use of brain imaging techniques to examine the brain structure has great potential to give timely, accurate and objective information about the relationships between nutritional interventions, growth, and neurodevelopment [11].

The impact of nutrition on brain development in preterm infants cannot be sufficiently emphasized. Early postnatal nutrient intake has been demonstrated to influence brain growth and maturation with the subsequent effects on neurodevelopment persisting into later childhood and adolescence. Appropriate nutrition could play a protective role against brain injury and development by its immunomodulatory and/or anti-inflammatory effects serving as a neuroprotective agent [10].

Published criteria for pediatric malnutrition has been historically intended for use in infants older than 37 weeks of corrected age and older than 30 days of age, and therefore, are not applicable to preterm infants. However, in 2018 Goldberg et al. recommended a set of indicators that can be used to identify malnutrition in the preterm neonatal populations [4]. These are appropriate for use from 2 weeks of life onwards given the expected postnatal diuresis resulting in a 7–20% weight loss in the first days of life. Once diuresis is complete, postnatal growth assessments such as weight and change in weight (z-score) become the primary indicators for evaluation of infant malnutrition [4]. Change in length (z-score) obtained from accurate length measurements is also a valuable indicator of malnutrition in preterm infants. The relationship between length, brain development, and neurodevelopmental outcome has been well established [1].

Our study revealed a positive association between malnutrition z-scores and total brain tissue volumes. As weight and length improved, brain tissue volumes increased. The indicators used to assess the nutritional status can differentiate between different degrees of malnutrition. Remarkably, all infants in this cohort were found to have some degree of malnutrition during their hospital stay, with the majority being moderate to severe. This study analyzed the maximum change in weight and length but did not analyze the duration of time an infant remained in a specific degree of malnutrition. When using maximum change in weight, most infants showed scores indicative of moderate malnutrition and when maximum change in length was assessed, most scores were consistent with moderate to severe malnutrition. Attempts have been made to enhance nutrition regimens; however, mixed results have been obtained in which increased protein and lipids could improve brain volumes as described by van Beek et al. [17], but in another study by Power et al., higher protein intake was not associated with brain size or neurodevelopmental outcomes [2]. Despite continuous efforts to optimize provision, length z-scores showed higher declines which could indicate that length is a more challenging parameter to improve in infants. Of note, studies have shown positive relationships between length and neurodevelopmental outcome [11,27,28,29]; this may be because lean mass deposition relies on adequate intakes of both protein and energy and linear growth reflects protein accretion and the structural growth of the brain [30].

Evidence has shown the mean total brain volume at term age is approximately 400,000 mm^3^, yet the largest volume obtained in this cohort was 281,400 mm^3^ [14]. All infants met the criteria for malnutrition and none of them met the suggested mean brain volumes in the literature at TEA. As shown in Table 2 and Table 3, nutritional deficiency seems to have more impact on infants’ total brain tissue volume with those born at a younger gestation and on SGA infants in comparison with those born at a later gestation and on those who were AGA. The SGA group had head sparing, meaning the infants had a birthweight less than the 10th percentile with normal head circumferences. Interestingly, the LGA group had smaller brain volumes than the AGA group; this could be explained by their greater degree of malnutrition, as reflected in a weight z-score decline of 1.9 (*p* < 0.05). However, this result is difficult to interpret due to the limited number of LGA infants in our cohort, *n* = 5. There was an upward trend in brain tissue volumes with greater gestational age at birth, but the infants born at 27 weeks showed lower brain volumes than the rest of the cohort, but these infants had higher degree of malnutrition than those at greater gestational ages. Regarding the associations with other clinical variables such as postnatal steroid use and *hs*PDA, our group was surprised by the lack of relationship with brain volumes, which is contrary to other studies [18,31,32,33,34]. The difference in findings in our study compared to others in the literature could be due to the heterogenous inclusion criteria or the different definitions used for long-term use of steroids and for *hs*PDA.

One of the strengths of our study was that our institution has a protocol where we perform term equivalent brain MRIs on all infants born less than 30 weeks around 36 weeks equivalent age. This study also utilizes a cohort covering a wide range of gestational ages from 23 weeks’ GA to 30 weeks’ GA, contributing to our understanding of early regional brain development.

One limitation was that this was a retrospective, single-center study that reported total brain tissue volume measurements with no differentiated tissue-specific brain regions as other groups have analyzed [3], potentially affecting the interpretation of the clinical outcomes. In future studies we hope to be able to include regional brain volumes. Studies have shown that the cerebellum is one of the areas that exhibit the highest growth rates [15,16]; however, cerebellums were not measured in this study, and we may have missed the effect of nutrition on this specific brain region. Further studies could include a larger patient population with finer-scale delineation of the distinct brain regions. Lastly, the quality of MRI manual segmentations may not precisely map the brain morphology of preterm infants. However, we consider that the total brain tissue volumes computed are sufficiently representative of underlying anatomy at TEA for the study participants and allow for feasible brain size associations with malnutrition.

## 5. Conclusions

Optimizing early nutritional support has been shown to improve neurodevelopment in preterm infants [35,36,37,38,39]. Recommended nutrient intake for preterm infants is not routinely achieved despite efforts to optimize provision. Nutrition is a modifiable factor and appropriately classifying preterm infants with malnutrition may increase awareness of suboptimal growth leading to long-term consequences. These results could serve as reference data for future timely nutritional interventions to promote optimal brain volume. The data found in our study was very encouraging and will serve as the foundation for future studies with a larger number of subjects and can be used to examine possible nutritional interventions to increase brain volumes. With the availability of in-unit MRI machines designed to image premature neonates, sequential MRI brain imaging may serve as a guide for nutrition management in a prospective manner thereby improving brain development and subsequent neurodevelopmental outcomes.

## Figures and Tables

**Figure 1 children-11-00121-f001:**
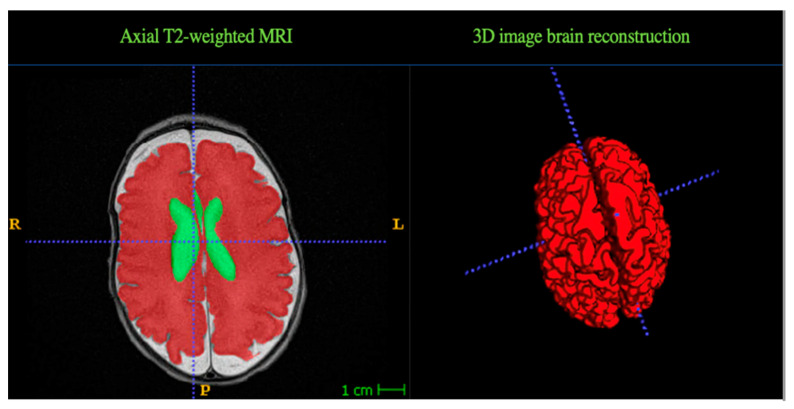
Segmentation results. Representative example of manual segmentation data obtained with ITK SNAP software, using the Mary Rutherford neonatal brain atlas. Two different brain structures are delineated in different colors. Green = Ventricles, Red = Brain tissue volume (grey and white matter).

**Figure 2 children-11-00121-f002:**
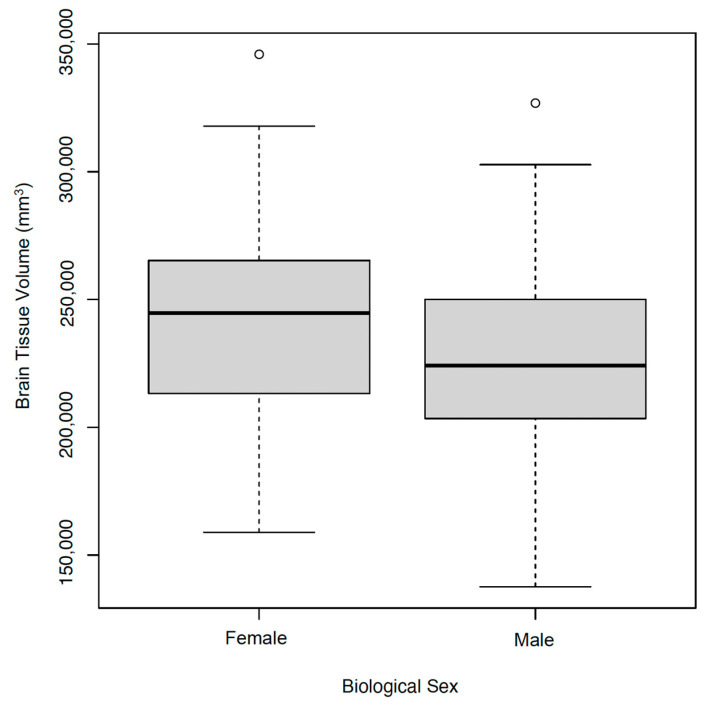
Brain Tissue Volume and Biological Sex. Smaller brain volumes were associated to the male gender.

**Figure 3 children-11-00121-f003:**
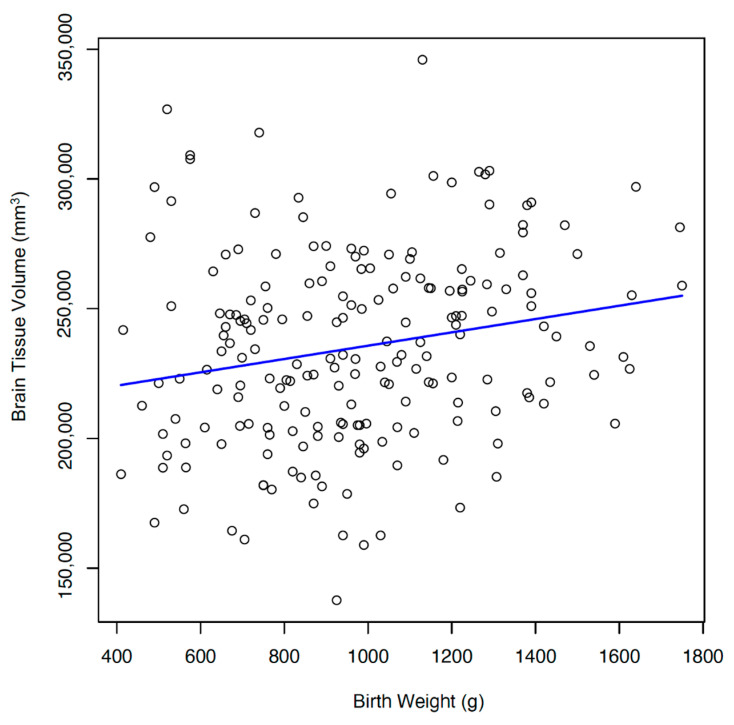
Brain Tissue Volume and Birth Weight. Significant positive association between birth weight and brain tissue volume. (r = 0.204).

**Table 1 children-11-00121-t001:** Baseline demographics and clinical characteristics.

Characteristic	Mean + SD	*n*	%
Gestational age at birth, wk.	26.8 ± 1.8	198	
Birth weight, g	973 + 294		
Male		103	52
Female		95	48
Brain volume, mm^3^	235,059 ± 37,045		
Max change in weight, z-score	−1.3 ± 0.6		
Max change in length, z-score	−1.5 ± 1.3		
SGA		22	11
AGA		171	86
LGA		5	2.5
IVH low grade		74	37
IVH high grade		13	7

Abbreviations: SGA, small for gestational age; AGA, appropriate for gestational age; and LGA, large for gestational age. IVH low grade: intraventricular hemorrhage grade 1 or 2; IVH high grade, intraventricular hemorrhage grade 3 or 4. SD = Standard deviation.

**Table 2 children-11-00121-t002:** Malnutrition scores and total brain tissue volumes.

Characteristic	23 wk. *n* = 10	24 wk. *n* = 17	25 wk. *n* = 19	26 wk. *n* = 29	27 wk. *n* = 34	28 wk. *n* = 47	29 wk. *n* = 42
Birth weight (g), mean (SD)	573 (115)	683 (100)	729 (125)	787 (178)	971 (171)	1128 (229)	1252 (259)
Max change weight(z-score), mean (SD)	−1.2 (0.8)	−1.7 (0.8)	−1.7 (0.8)	−1.3 (0.7)	−1.4 (0.6)	−1.2 (0.6)	−1.1 (0.5)
Max change length (z-score), mean (SD)	−1.8 (1.0)	−1.9 (0.9)	−2.0 (0.7)	−1.2 (2.7)	−1.7 (0.8)	−1.5 (0.8)	−1.1 (1.0)
Total brain volume (mm^3^), mean (SD)	234,350 (43,762)	231,847 (25,145)	244,605 (38,345)	239,696 (38,767)	205,882 (38,136)	242,304 (29,398)	244,519 (34,573)

Abbreviations: wk., week; *n*, number; and (SD), standard deviation.

**Table 3 children-11-00121-t003:** Malnutrition scores and total brain volumes per size at birth.

Size	*n*	Brain Volume (mm^3^), Mean (SD)	Max Change Weight, (z-Score), Mean (SD)	Max Change Length, (z-Score), Mean (SD)
SGA	22	227,595.5 (44,160)	−1.3 (1.0)	−1.6 (1.1)
AGA	17	1236,135.1 (36,179)	−1.3 (0.6)	−1.5 (1.4)
LGA	5	231,100.0 (36,708)	−1.9 (0.2)	−1.4 (0.1)

Abbreviations: SGA, small for gestational age; AGA, appropriate for gestational age; LGA, large for gestational age; *n*: number; and (SD), standard deviation.

## Data Availability

The data presented in this study are available in article.

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
