# Peer review of "Impact of Nutritional Status on Total Brain Tissue Volumes in Preterm Infants"

_children, 2024, doi:10.3390/children11010121_

Round 1

Reviewer 1 Report

Comments and Suggestions for Authors

The authors describe a well-performed study studying the effect of nutrition on term equivalent age brain volumes on MRI. Few corrections/changes are recommended before the paper can be considered for acceptance.

Major:

1. Lines 264-266: As shown in Tables 3 and 4, nutritional deficiency seems to have more impact on infant’s total brain tissue volume on those born at younger gestation and on SGA infants in comparison with those born at later gestation and on those who were AGA.

Was there statistically significant difference between the various gestational ages with respect to their nutritional deficiency and brain volume??

2. Lines 268-70: Interestingly, the LGA group had smaller brain volumes than the AGA group, this could be explained by their greater degree of malnutrition, as reflected in a weight z-score decline of 1.9. 

Again, please state if this difference was statistically significant.

3. The authors conclude that "Neonatal MR-based brain tissue volumes of preterm infants could serve as objective, quantitative and reproducible surrogate parameters of early brain development."-- however, this is not supported by the data of this study, please delete/ modify.

4. Additionally, various other studies have looked at nutrition and association with term equivalent age brain volumes in preterm infants. Some are referenced below. These studies must be discussed and cited.

Power, Victoria A., et al. "Nutrition, growth, brain volume, and neurodevelopment in very preterm children." The Journal of Pediatrics 215 (2019): 50-55. 

van Beek, Pauline E., et al. "Increase in brain volumes after implementation of a nutrition regimen in infants born extremely preterm." The Journal of Pediatrics 223 (2020): 57-63.

Bell, Katherine Ann, et al. "Associations of body composition with regional brain volumes and white matter microstructure in very preterm infants." Archives of Disease in Childhood-Fetal and Neonatal Edition 107.5 (2022): 533-538.

Minor: Line 169-177: The paragraph and the table 2 are repetitive. One of the two must be deleted

Author Response

Reviewer #1

1. Lines 264-266: As shown in Tables 3 and 4, nutritional deficiency seems to have more impact on infant’s total brain tissue volume on those born at younger gestation and on SGA infants in comparison with those born at later gestation and on those who were AGA. Was there statistically significant difference between the various gestational ages with respect to their nutritional deficiency and brain volume??

Response:  There is statistical difference between gestational ages and length z-score and brain volumes but not when analyzing gestational ages with weight z-score and brain volumes.

2. Lines 268-70: Interestingly, the LGA group had smaller brain volumes than the AGA group, this could be explained by their greater degree of malnutrition, as reflected in a weight z-score decline of 1.9. Again, please state if this difference was statistically significant.

Response: Yes, there is statistically significant difference in the LGA group, and the p value has been added to the manuscript, it can be found in lines 290-291.

3. The authors conclude that "Neonatal MR-based brain tissue volumes of preterm infants could serve as objective, quantitative and reproducible surrogate parameters of early brain development."-- however, this is not supported by the data of this study, please delete/ modify.

Response: We have made the suggested changes and we removed the statement.

4. Additionally, various other studies have looked at nutrition and association with term equivalent age brain volumes in preterm infants. Some are referenced below. These studies must be discussed and cited.

Response: We have added the suggested references, and these are included in the discussion section.

  1. The paragraph and the table 2 are repetitive. One of the two must be deleted.

Response: We have made the suggested modifications.

Reviewer 2 Report

Comments and Suggestions for Authors

This is a very interesting paper and I must admit that as a neonatologist I fully understand the need for appropriate feeding for preterm babies.

Nutrition is very important, starting with the respiratory pathologies that appear in the first ours of life. Also, as the preemies grow, it becomes more obvious that the calories are essential in their development and the brain ends up suffering many times because of scarce nutrients.

Followup of the preemies can emphasize the correct medical treatment received by premature infants from the very beginning of their life. 

Congratulations!

Comments on the Quality of English Language

English is fine

Author Response

Thank you very much for taking the time to review our manuscript. We didn't find any specific or detailed suggestions/modifications to be made.  We greatly appreciate your comments. We will continue to look for the other reviewers comments as well.

Reviewer 3 Report

Comments and Suggestions for Authors

Thank you for forwarding the article for review. Here are some comments that you should consider 

1. please accurately present the inclusion criteria for the study

2. please consider shortening the conclusions. In their current form they are too long and are a repetition of the results, perhaps they can be generalized and shortened

3 I think that the literature should be expanded.

4.The work is interesting.

Author Response

Reviewer #3

Response to reviewers from the Children Journal

           Thank you very much for taking the time to review this manuscript. Please find the detailed responses below and the corresponding revisions/corrections highlighted.  

  1. Please accurately present the inclusion criteria for the study.

Response: Thank you for your suggestion, this has now been added to the manuscript in the lines 84-85.

  1. Please consider shortening the conclusions. In their current form they are too long and are a repetition of the results, perhaps they can be generalized and shortened.

Response: Thank you for pointing this out. We have made modifications to the conclusions to make more concise by removing the results.

3 I think that the literature should be expanded.

Response: we have made the suggested additions and can be found highlighted throughout the manuscript.

4.The work is interesting.

Response: Thank you very much.

Reviewer 4 Report

Comments and Suggestions for Authors

I thank the authors for investigating this very important area.

This single centre retrospective chart review investigated if nutritional status in preterm infants is associated with brain tissue volumes at term equivalent age using malnutrition guidelines for preterm infants. The relationship between MRI based brain tissue volumes were analysed with maximum weight and length z-scores and found a significant linear association.

Abstract:

Well written and succinct.

Introduction:

1.       Kindly insert references for lines 37 to 40 and for line 45.

Methods:

1.       Line 100; it looks like there is a word missing.

2.       In relation to PDA, were there any other echocardiographic parameters other than ductal diameter that was used. Treatment itself can be subjective. Was feed increment cautious in infants who had features of ductal steal. Please discuss.

3.       Which babies had imaging on 1.5T and which on 3T? Who made this distinction. Please discuss.

4.       Please insert reference for line 125.

5.       What is the nutritional practice for this unit. I think it would be very useful for the reader to understand this.

6.       What is the rationale for only using weight and length for brain tissue volumes. The obvious question that the reader will have is what is the relationship of head circumference measurements (an easily performed clinical measurement). There seems to be no mention of this in the paper. Please discuss why head circumference was not used in assessing brain tissue volumes especially when you have access to the data.

7.       It would be really useful to understand how head circumference z-scores correlated with brain tissue volumes. This would make it more generalisable especially for unit that do not perform routine MRI brain at 36 weeks.

8.       How did the team account for the intra and inter observer variability – more so for length measurements?

Results:

1.       Please represent the study patient flow using a chart. It is very difficult to follow this in its current form. Also please give details about 22-week gestation infants.

2.       I am unsure how much information figure 1 is providing to the reader. Do you need this?

3.       Lines 169 to 171 – replication of data. This is already presented in the table.

4.       Please provide graphical data on brain tissue volumes plotted against head circumference.

5.       It would be good to provide the exact p-value.

6.       Please provide ethnicity details of the study participants.

Discussion:

1.       Please discuss why postnatal steroid use of only >1 week was used. It would be good to describe unit policy about postnatal steroid use. Also is this steroid use for chronic lung disease or hypotension management. Please elaborate

2.       Please discuss why the weight and length z-scores were chosen as a parameter for this study. Why not head circumference z-scores.

3.       Please discuss why there was no association found between brain tissue volumes and type of enteral feeds.

4.       Which infants received donor breast milk.

5.       Why were male infants found to have lower brain tissue volumes

6.       The lack of a relationship between brain volumes and postnatal steroid use and haemodynamically significant PDA seen in this study is different to the findings from other studies. Please discuss why this is the case

Comments on the Quality of English Language

Good command of English.

Author Response

Reviewer #4

Response to reviewers from the Children Journal

           Thank you very much for taking the time to review this manuscript. Please find the detailed responses below and the corresponding revisions/corrections highlighted in the manuscript.  

Abstract:

Well written and succinct.

Introduction:

  1. Kindly insert references for lines 37 to 40 and for line 45.

Response: We agree. We have added the references for the above-mentioned lines.

Methods:

  1. Line 100; it looks like there is a word missing.

Response: - thank you, we have rewritten this sentence.

  1. In relation to PDA, were there any other echocardiographic parameters other than ductal diameter that was used. Treatment itself can be subjective. Was feed increment cautious in infants who had features of ductal steal. Please discuss.

Response:  Thank you for pointing this out, indeed treatment itself can be subjective. The patients who were treated did show clinical signs of ductal steal or pulmonary congestion thus were treated, however not all babies had an echo done. And yes, fluid, or cautious feeding increment is part of their management. However, we didn’t find statistically significant clinical correlation between a PDA as defined in our paper and malnutrition, therefore we didn’t analyze the feeding volumes during treatment.  We have added this explanation to the manuscript to further clarify.

  1. Which babies had imaging on 1.5T and which on 3T? Who made this distinction. Please discuss.

Response: This decision is made at the radiologist’s discretion. We didn’t collect information regarding which babies had 1.5T or 3T. We do not feel the different magnetic field strengths would’ve made a significant difference given we analyzed total brain volumes as opposed to individual brain structures. In addition, manual segmentation was performed, we didn’t use an automated program which would’ve been affected by magnetic field strength.

  1. Please insert reference for line 125.

      Response: Reference has been inserted.

  1. What is the nutritional practice for this unit. I think it would be very useful for the reader to understand this.

Response: We agree with this suggestion. We have briefly described the unit nutritional practice in our manuscript; this is an extensive guideline, but we believe we have added what’s pertinent to the manuscript. This can be found under the subsection 2.4 Nutritional status.

  1. What is the rationale for only using weight and length for brain tissue volumes. The obvious question that the reader will have is what is the relationship of head circumference measurements (an easily performed clinical measurement). There seems to be no mention of this in the paper. Please discuss why head circumference was not used in assessing brain tissue volumes especially when you have access to the data.

Response: Thank you for touching on this subject. We did have access to this data and admit there is no mention of the head circumference measurement given that we are focusing on the primary indicators of neonatal malnutrition described by Goldberg et al. Identifying malnutrition in preterm and neonatal populations: Recommended indicators. J Acad Nutr Diet 2018. These indicators take into consideration mostly weight and length z-scores, weight gain velocity, nutrient intake, and a few others but not head circumference. We chose to use weight and length z-scores considering its data we had readily available and its generalizable.

  1. It would be really useful to understand how head circumference z-scores correlated with brain tissue volumes. This would make it more generalizable especially for unit that do not perform routine MRI brain at 36 weeks.

Response: We agree with your comment. It would be interesting to review head circumference with brain volumes however, we were looking into associations between malnutrition indicators and brain volumes, and as described in the question above, head circumference is not included as a neonatal malnutrition indicator.

  1. How did the team account for the intra and inter observer variability – more so for length measurements?

Response: We made an effort to describe the procedure of length measurements in our unit. This is described in the subsection 2.4 Nutritional Status. Length is measured by placing the infant on their back in the center of a length board, measuring to the nearest 0.1 cm. This has been standardized in our unit to minimize variability.

Results:

  1. Please represent the study patient flow using a chart. It is very difficult to follow this in its current form. Also please give details about 22-week gestation infants.

Response: Agree. We have added a flow chart to clarify the study population and gave details regarding the 22-week infants.

  1. I am unsure how much information figure 1 is providing to the reader. Do you need this?

Response: Figure 1 is a representation of the brain volumes obtained after manual segmentation. We consider important for this figure to be present in the manuscript to illustrate the delineated structures in the brain MRI and to show the reader the type of software used.  

  1. Lines 169 to 171 – replication of data. This is already presented in the table.

Response: Agree. We have made the suggested changes. We deleted the paragraph and left all the information in the table.

  1. Please provide graphical data on brain tissue volumes plotted against head circumference.

Response: Although we do agree head circumference is very valuable to investigate, the objective of this study was to assess the relationship between malnutrition and brain volumes. Other interesting studies have found positive associations between brain size/volumes and head circumference.

  1. It would be good to provide the exact p-value.

Response: Please see the above response.

  1. Please provide ethnicity details of the study participants.

Response: Unfortunately, no ethnicity data was collected.

Discussion:

  1. Please discuss why postnatal steroid use of only >1 week was used. It would be good to describe unit policy about postnatal steroid use. Also is this steroid use for chronic lung disease or hypotension management. Please elaborate.

Response: Agree. Steroid use for > 1 week was chosen given generally steroids are prescribed for roughly 1-2 weeks or more (in the case of chronic lung disease). Some infants receive spot steroid doses for hypotension and those were not included, given their short-term use. No distinction was made for the reasons why an infant received steroids. (Whether it was for chronic lung disease, hypotension, or other anti-inflammatory purposes). Considering in our unit, steroid tapers are indicated if an infant received systemic steroids for > 1 week or >40 mg of prednisone per day, we chose arbitrarily to define postnatal steroid use as such. There is no current steroid policy in our unit, they are prescribed at the discretion of the treating physician.

  1. Please discuss why the weight and length z-scores were chosen as a parameter for this study. Why not head circumference z-scores.

      Response: Thank you for this valuable comment, we have discussed this in the ‘nutritional status’ section in lines 139-144 and in the Discussion section in lines 249-260.

  1. Please discuss why there was no association found between brain tissue volumes and type of enteral feeds.

Response: After running the analysis of the interaction between the different types of enteral feeds (breastmilk, donor breast milk or formula), no statistical significance was found. 

  1. Which infants received donor breast milk.

Response:  a total of 22 out of the 198 infants received donor breast milk. This data was collected in percentages throughout the entirety of their NICU stay. For example, one infant may have received breastmilk 20%, donor breastmilk 40% and formula 40%. No differences were found between brain volumes and type of enteral feed; thus we didn’t delve into details.

  1. Why were male infants found to have lower brain tissue volumes?

Response: This was an interesting observation. We are uncertain of the etiology and do not want to speculate, however, this can be investigated in the future in a larger study.

  1. The lack of a relationship between brain volumes and postnatal steroid use and haemodynamically significant PDA seen in this study is different to the findings from other studies. Please discuss why this is the case.

Response: We agree with your statement, and it was also a surprise for our group. Offering an explanation would be pure speculation, and we can only state the results we found.  This being said, it could be that the inclusion criteria and definitions for a hemodynamically significant PDA or long-term steroid use are different from one study to the other.  For instance, a paper by Lemmers et al., in 2016 from the journal of Pediatrics tilted ‘Patent Ductus Arteriosus and Brain Volume’, defines a hPDA with much more finer and detailed criteria (left atrial or left ventricular dilatation 1:4, internal ductal diameter >1.4 mm/kg, and left pulmonary artery end diastolic flow >0.2 m/second).

Comments on the Quality of English Language

Good command of English.

Round 2

Reviewer 4 Report

Comments and Suggestions for Authors

Thank you for addressing all the questions and well done on this research in this important area.